# When Body Art Goes Awry—Severe Systemic Allergic Reaction to Red Ink Tattoo Requiring Surgical Treatment

**DOI:** 10.3390/ijerph191710741

**Published:** 2022-08-29

**Authors:** Agata Szulia, Bogusław Antoszewski, Tomasz Zawadzki, Anna Kasielska-Trojan

**Affiliations:** 1The Military Medical Faculty, Medical University of Lodz, 90-419 Lodz, Poland; 2Plastic, Reconstructive and Aesthetic Surgery Clinic, Medical University of Lodz, Kopcinskiego 22, 90-153 Lodz, Poland

**Keywords:** tattoo, tattoo complication, hypersensitivity reaction, case report, Poland, hygiene and public health, health promotion

## Abstract

The aim of this report is to present a case of a patient who developed unusual systemic hypersensitivity reaction to a red-pigmented tattoo and to discuss diagnostic difficulties in case of systemic reactions to tattoo ink. The patient reported erythroderma on his arms and chest accompanied by plaque elevation of red parts of his most recently performed forearm tattoo as his primary symptoms. His health condition entailed a prolonged topical and intravenous immunosuppressive therapy, which proved ineffective. Over a year after emergence of initial symptoms, he presented to the Plastic Surgery Clinic with generalized erythroderma, systemic lymphadenopathy, elevation and granuloma formation in red tattoos on his forearm and complaints of fatigue and inability to participate fully in work-related and social activities. The patient underwent six staged excisions with direct closures, flap plasties and full-thickness skin grafts. Following completion of each surgical resection, the patient’s symptoms gradually subsided. We find this case illustrative of a clinical challenge that delayed hypersensitivity reactions to red tattoos may pose. Furthermore, we provide insights on management of hypersensitivity reactions. This report underlines the importance of social awareness of and public health approach to tattoo complications as key to successful prevention, identification and treatment of adverse reactions to tattoos.

## 1. Introduction

Tattooing is a millennia-old form of body modification performed by depositing pigment in the dermis layer of the skin. It may be perceived simply as a decorative marking on the body or serve as a way to enhance one’s sense of individuality, increase attractiveness or express affiliation to a subculture [1]. The number of tattooed people has been significantly increasing and, subsequently, so has the number of tattoo-related complications [2]. As any foreign body introduced into the skin, tattoo ink creates a risk for a broad range of adverse events: inter alia photosensitivity, infection, trauma related to needle insertion or allergic response to the pigment [3,4]. Some studies estimate that 6% to 8% of tattooed people are affected by a tattoo complication which requires professional help [5]. Diversity of adverse tattoo reactions may be attributed to the color of tattoo ink used [2]. Black tattoos are generally considered safe, perhaps because of their natural origin, and relatively rarely cause complications [6]. Colorful tattoos, made with green, yellow, gold and red pigment owe their vibrancy to azo pigments and, nowadays largely omitted, metals, which may precipitate hypersensitivity reactions exhibiting as ulcers, plaques or eczematous lesions [7,8,9]. Regardless of the pigment used, all chronic tattoo sequelae reduce quality of life and are a significant burden to suffering patients [10,11]. In particular, red ink is frequently described as the culprit of adverse tattoo reactions [4]. The irritating potential of the red pigment might be explained by its origin—even though red ink is now rarely produced using highly allergenic or even toxic metal compounds, and is usually obtained from safer for the skin, primarily organic compounds, such as azo dyes and quinacridone, the color still carries a risk of precipitating delayed hypersensitivity reaction [12]. Additionally, as any tattoo ink, red pigment might be contaminated by various chemicals, toxins, pathogens and other allergens, all of which might prompt allergic complications. Usually, delayed hypersensitivity reaction precipitated by red pigment emerges some time after uncomplicated healing. It typically manifests as plaque elevation, granulomatous reaction, extensive hyperkeratosis or ulcerous necrotic change of red-tattooed skin and is caused by pigment-associated allergic sensitization [13]. Patients report symptoms of swelling and itching in the affected skin area. Local corticosteroid treatment usually does not yield convincing results [14]. Laser treatment is contraindicated as it might amplify allergic reaction and potentially precipitate anaphylaxis because of photochemical breakdown of pigment upon laser light exposure [13,15,16]. In cases where noninvasive treatment fails to provide relief of symptoms, surgical intervention is suggested [14]. Herein, we report an unusual case of a patient with allergic tattoo reaction which presented originally as erythroderma of the chest and arms with slight plaque elevation in red parts of the patient’s forearm tattoos two months after their completion. As such a manifestation of allergic tattoo reaction is not common, and given that the symptoms occurred in the setting of the first wave of SARS-CoV-2 pandemic, the patient underwent a long diagnostic and therapeutic process before arriving at the right diagnosis and treatment, which comprised multiple staged surgical interventions and different reconstructive techniques.

The aim of this report is to present a case of a patient who developed unusual systemic hypersensitivity reaction to a red-pigmented tattoo and to discuss diagnostic difficulties in case of systemic reactions to tattoo ink.

## 2. Case Presentation

### 2.1. Clinical Presentation

A 36-year-old Caucasian male was referred to the Plastic Surgery Clinic by an allergist. He presented with skin changes (edema and granulomas demarcated by red floral- and flame-shaped tattoos) on red parts of tattoos on his right forearm and complained of fatigue and reduced quality of life. The patient had many extensive colorful tattoos on his chest and both upper limbs, which were performed a few years earlier, except for red, black and green tattoos on his right forearm, which were performed about 6 weeks prior to the emergence of systemic erythroderma and cutaneous changes (Figure 1 and Figure 2). He had no previous history of adverse reactions to tattoos performed earlier. All of the patient’s tattoos were performed in the same professional tattoo salon; however, the exact composition of the red ink used was not known (no information about the manufacturer was available). Upon his admission to the Clinic, apart from systemic erythroderma with general anhidrosis and skin changes, also of note was loss of all of the patient’s scalp and body hair. At the time of presentation, he had been receiving oral methylprednisolone (16 mg daily), oral leukotriene receptor antagonist (10 mg daily) and oral rupatadine (10 mg daily). He also reported the symptoms and side effects of received treatment had significantly reduced his ability to participate in work- and social-related duties and impaired his quality of life.

The patient’s medical history includes Hashimoto’s thyroiditis treated with levothyroxine, a period of self-administered testosterone misuse for muscle-mass building purposes, and hospitalization for pneumonia associated with COVID-19 infection during immunosuppressive treatment of erythroderma and skin changes. His family history includes hypothyroidism. He has no known history or family history of autoimmune conditions, such as sarcoidosis or dermatologic diseases.

### 2.2. Laboratory Investigations

Bloodwork revealed mildly elevated C-reactive protein (CRP) (8.9 mg/L) and eosinophilia (2.4 × 103/μL, 31.7%). Diagnostic tests that the patient underwent also included biopsy of an enlarged lymph node from his groin, which showed deposits of red pigment and lymphocytic infiltration. He also underwent multiple skin biopsies of the reactive tattoo area and nontattooed skin affected by pruritis at various stages of treatment. Nontattooed skin specimens yielded nonspecific findings, while the tattoo bioptates showed pigment deposits with intensive inflammatory cell infiltration and erosion throughout the epidermis and skin appendages. Biopsies did not show signs of lymphoproliferative diseases. He tested negatively for HIV, HBV, HCV, parasites, and did not have any remarkable findings in computer tomography imaging of the chest, abdomen and pelvis. Patch testing for metal allergy yielded negative results.

### 2.3. Differential Diagnosis and Clinical Course

Until the onset of skin lesions, the patient was a semiprofessional athlete, maintained good health condition and was subject to regular medical supervision. Medical interview revealed that the first symptom—erythroderma—appeared slightly over 1.5 months after tattoos on his left forearm were made and coincided with the time when the patient was delegated to work in a COVID-19 hospital as a professional soldier assisting medical staff. It was at the time of the first wave of the SARS-CoV-2 pandemic, before vaccinations against the disease were available. The position required him to wear protective clothing and expose his skin to substantial amounts of disinfectants, which had an irritating effect and caused outbreaks on the skin. Initial erythema on the forearms and torso progressed into painful erythematous, populous and squamous lesions on his head, neck, torso and limbs with pruritis, and hyperkeratosis of the soles of the feet and hands. Associating skin changes with the nature of his work prompted the patient to undertake testing for COVID-19, which yielded negative results. The patient resigned from work at the COVID-19 hospital to limit his exposure to irritants, yet the symptoms did not resolve. Over the course of the year, he looked for treatment at dermatology, internal medicine, hematology, and allergology departments to relieve his symptoms. Given the untypical, challenging clinical presentation and emergence during the COVID-19 pandemic, the diagnosis of chronic allergic tattoo reaction was not an easy one to reach, and, thus, multiple diagnoses were suspected, including atopic dermatitis, pityriasis rubra pilaris, follicular mucinosis and lymphoma.

### 2.4. Treatment and Interventions

The patient underwent several courses of conservative treatment, which included oral and intravenous immunosuppressive therapy (dexamethasone, hydrocortisone, prednisone, methylprednisolone, azithromycin, cyclosporin A and methotrexate), UVA1 phototherapy, retinoids, antibiotics, protective and lubricating skin ointments, and oral antihistamine drugs. The symptoms partially improved on high doses of intravenous immunosuppressants but recurred with each attempt to reduce the dose. After 10 months of treatment with only minimal effects, an allergist managing the patient grew suspicious of the appearance of the tattooed skin on the patient’s right forearm and considered his symptoms might be attributed to hypersensitivity reaction to red pigment in the ink used to perform one of the tattoos. Due to his working diagnosis, he referred the patient to our Plastic Surgery Clinic, where, after examination of the patient and his medical history, the patient was scheduled for surgical excision of the tattooed skin of his left forearm (Figure 3). Steroids were discontinued at the beginning of surgical treatment.

### 2.5. Outcomes

The result of surgical intervention was a complete excision of all reactive parts of the tattoos within the course of six surgeries performed in approximately one-month intervals. The resection was accomplished with the use of direct closures with presuturing, local flap plasties and full-thickness skin grafts harvested from the patient’s groin area. After each excision session, the healing period was uneventful and, with each resection performed, the patient’s systemic symptoms improved, with a significant relief of the symptoms following final resection. At a follow-up visit, three months after completion of surgical treatment, the patient’s recovery had been progressing with no recurrence of erythroderma (Figure 4). Moreover, his blood tests normalized (eosinophil count 0.62 × 10^3^/μL, 13%).

## 3. Discussion

Tattoos, as any form of introducing a foreign body into the skin, might cause adverse reactions. The response might be of diverse nature and, aside from fairly easy to manage acute complications, such as photosensitivity or infection, common and problematic in their nature are chronic tattoo allergic reactions. As in the case of our patient, the major culprit of delayed allergic response to tattoo is red ink. Historically, red pigment was largely produced with the use of metals, including highly toxic mercury derivatives. These have now been replaced by less dangerous cadmium and ferric compounds, as well as organic substances (e.g., brazilwood and sandalwood) [12]. Despite the change, reactions to red tattoos continue to occur. This might be attributed to the composition of currently used tattooing substances, which still carry some allergenic potential, as well as unregulated, untested additives to tattoo inks, which might also possess allergenic potency. As the allergen is usually unidentified, the exact pathomechanism of chronic allergic reaction to red ink tattoo is also not fully explained. Presumably, the mechanism is that metabolism of the ink or reaction of an ink antigen with a carrier protein in the dermis over a long time triggers a delayed hypersensitivity reaction [17,18]. In line with the presumed pathophysiology, chronic allergic tattoo reactions occur months or years following body art completion. They typically present as, localized to the red tattoo area, swelling, granulomas, ulceration, pruritis or hyperkeratosis [4,5,13]. Very rarely, systemic response, as in the case of our patient, occurs [13,19]. It may manifest as widespread dermatitic eruption, rosacea, scaling, erythema and papules on extensive parts of the body. In Table 1, we presented a review of clinical cases concerning complications of red tattoo ink and we have suggested a diagnostic algorithm based on the literature and on our clinical experience for lesions suspected of being related to tattoo ink (Figure 5).

Treatment in allergic tattoo reactions typically starts with topical, oral or intralesional corticosteroids and antihistamines. Sometimes laser removal is taken into consideration. These methods, however, are often ineffective in treatment of inflamed, thickened dermis in pigment-laden, reactive tattoos [13]. In particular, laser removal is contraindicated, as there are concerns that it may induce photochemical changes of the pigment and produce new toxic chemicals and novel allergens [14]. In such cases of reactive tattoos, a logical course of action seems to be surgical intervention. Sepehri et al. proposed dermabrasion and dermatome shaving as the first-line treatment for chronic allergy reactions in tattoos. These techniques provide complete excision of the reactive tissue and symptom relief; however, they are usually performed for removal of superficially located ink, as deeper shaving may entail prolonged healing and major scarring [37]. In the case of our patient, in whom the reaction was reaching deep into the dermis and affected a large surface of the skin, full-thickness excision with direct closures and skin grafting seemed sensible [38]. Removal of the dermis enabled complete elimination of the pigment, while full-thickness skin grafts harvested from the patient’s groin allowed for replacing large voids of the skin with an acceptable aesthetic effect. The treatment relieved the patient of cumbersome symptoms. Furthermore, he also reported improved quality of life. It is in line with studies [10,11] assessing the impact of chronic tattoo reactions on the quality of life, which conclude these may significantly impair the quality of patients’ life and should be given priority attention and qualified treatment. While deciding on the mode of treatment in our Plastic Surgery Clinic, improving the patient’s self-perception and psychological well-being was an important aspect, as the patient reported being unable to fully participate in work and private life activities due to the allergic reaction and adverse effects of long and ineffective treatment.

## 4. Conclusions

Tattoo ink is a frequent cause of delayed hypersensitivity reaction and potential serious sequelae should be taken into consideration when deciding on getting a tattoo, particularly made with red ink. Our case is illustrative of serious and untypical complications red tattoos may give rise to. Care should be taken upon examination of systemic skin changes in tattooed patients, as failure to reach a proper diagnosis puts them at risk of prolonged therapy and its side effects, and may significantly impair their quality of life. Upon detection of a chronic allergic tattoo reaction, surgical excision should be taken into consideration as an effective symptom-relieving treatment. Additionally, the report underlines the importance of social and clinical awareness of tattoo complications, especially in the era of rising popularity of tattooing.

## Figures and Tables

**Figure 1 ijerph-19-10741-f001:**
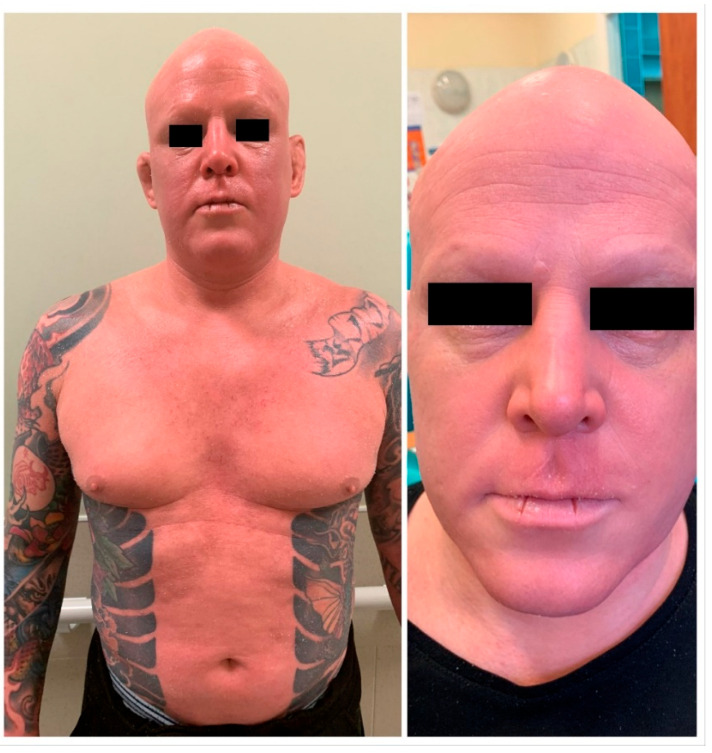
Systemic erythroderma, skin dryness and discoloration, loss of skin appendages (hair and sebaceous glands) on patient presentation in plastic surgery out-patient clinic.

**Figure 2 ijerph-19-10741-f002:**
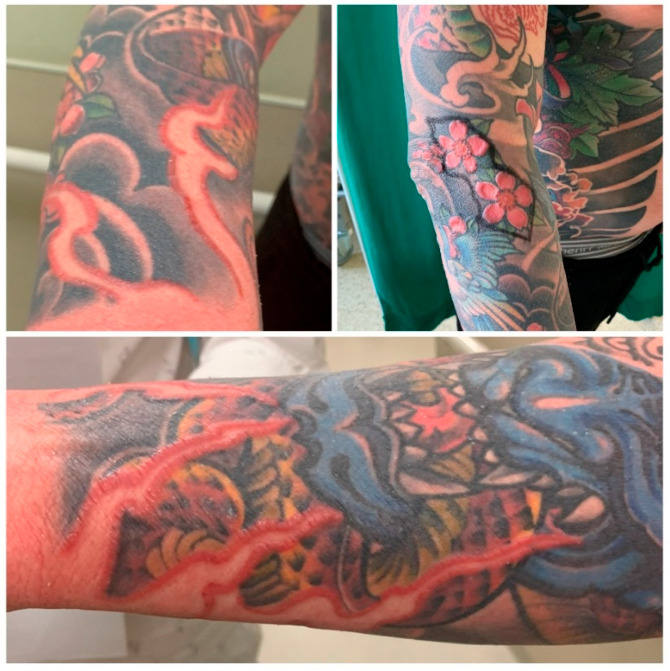
Skin changes (edema and granulomas demarcated by red floral- and flame-shaped tattoos) on red parts of tattoos on patient’s right forearm.

**Figure 3 ijerph-19-10741-f003:**
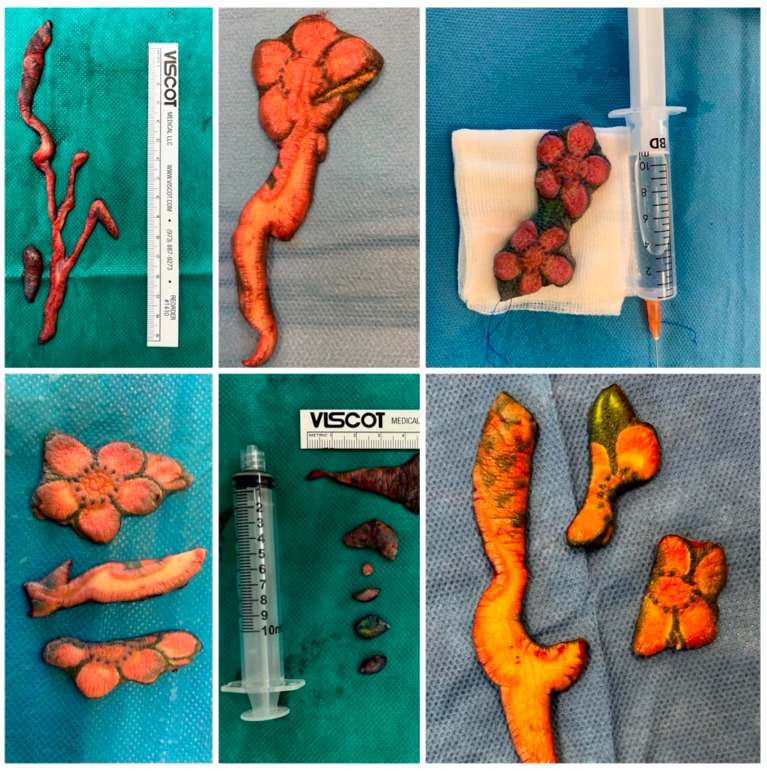
Excised parts of the tattoo of the right forearm with skin changes.

**Figure 4 ijerph-19-10741-f004:**
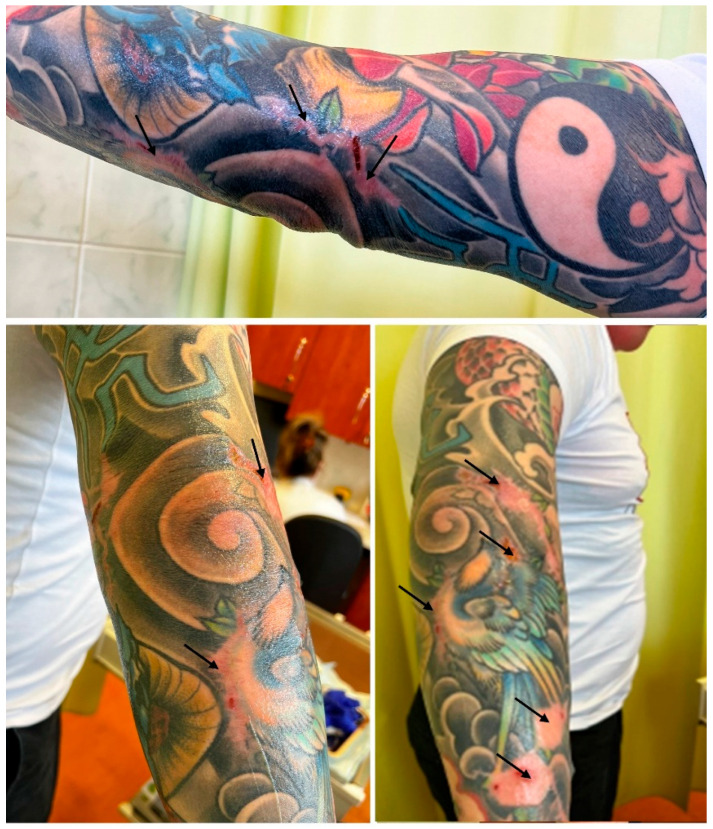
Patient’s forearm after staged excision of the red-pigmented parts of the tattoo (arrows—skin grafts and scars), remission of skin erythroderma and skin changes (3 months after completion of treatment).

**Figure 5 ijerph-19-10741-f005:**
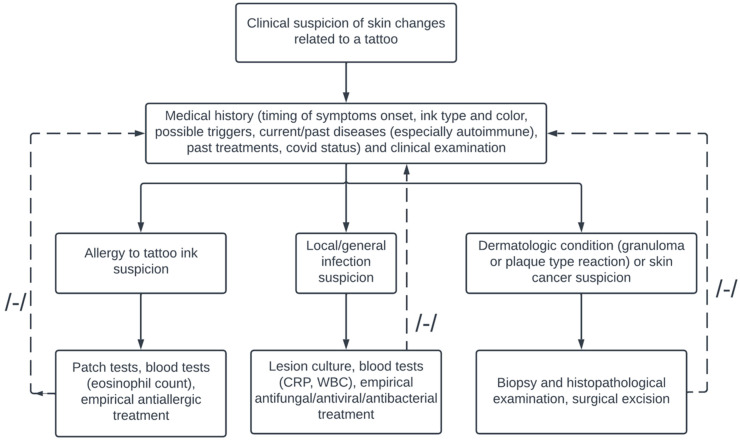
Diagnostic algorithm in case of lesions suspected of being related to a tattoo.

**Table 1 ijerph-19-10741-t001:** Review of clinical cases concerning red ink tattoo complications from the last decade.

Reference	No. of Cases, Sex, Age	Clinical Appearance	Onset of Symptoms	Tattoo Site	Symptom Localization	Investigations	Treatment and Outcome
McPhie et al. (2021) [18]	1, M, 51	Hyperkeratotic, erythematous nodules and raised, proliferative lesions	14 months	Dorsum of right hand	Confined to tattoos	Biopsy reactive atypia, a delayed-type hypersensitivity reaction	Clobetasol 0.05%, excision of hand nodules
Gómez Torrijos et al. (2021) [20]	1, male (M), 22	Itching, purpuric, pruritic papules	1 month	Left forearm	Confined to tattoo	Biopsy: eczema Patch tests: (+) cobalt chloride, (+) red tattoo ink in water and petrolatum	No data
Van Rooij et al. (2020) [21]	1, M, 48	Multiple squamoproliferative, erythematous lesions	12 years	Right dorsal foot	Confined to tattoo	Biopsy eruptive keratoacanthomas Culture: (−)	Patient refused treatment
Badavanis et al. (2019) [22]	1, female (F), 30	Verruciform skin erosion, nodule	3 years	Left ankle	Confined to tattoo	Culture: (−) X-ray of the leg (−) Biopsy: epidermal pseudoepitheliomatous hyperplasia	Excision
Sauvageau et al. (2019) [23]	1, M, 73	Verrucous growth	10 years	Left knee	Confined to tattoo	Biopsy: pseudoepitheliomatous hyperplasia	Shave removal
Price et al. (2018) [24]	1, F, 40	Pruritic, papulonodular eruptions	1 month	Right foot, right arm	On and near red tattoos	Patch tests: (−) Biopsy: allergic contact dermatitis	Fluocinonide (topical), oral prednisone, fractional laser (ineffective), Excision and mycophenolate mofetil (effective)
Saunders et al. (2018) [25]	1, F, 28	Firm plaque and papule	About 1 year	Left arm	Confined to tattoo	Biopsy: melanocytic nevus	No data
King et al. (2017) [26]	1, M, 46	Verruciform skin nodules on shin red tattoo followed by induration and itchiness of previous red forearm tattoo	2 months	Shin and forearm	Confined to tattoos	Biopsy atypical epidermotropic infiltrate of hyperchromatic lymphocytes, variable degree of verruciform acanthosis and hyperkeratosis	Excision and intralesional steroids (effective)
Wambier et al. (2017) [27]	1, F, 35	Pruritic nodules	No data	Right leg, ankle, foot, trunk	Confined to tattoos	Lab tests (−) Biopsy: epidermal hyperplasia, dermal fibrosis, chronic, lymphoplasmacytic inflammatory infiltrate around ink deposits	hydroxyzine topical clobetazol 0.05%, fludroxycortide (ineffective), Intralesional tramcinolone acetate (effective)
Sherif et al. (2017) [28]	1, F, 54	Nodular, erythematous plaque	8 years	Right lower leg	Confined to tattoo	Biopsy: epidermal acanthosis and hyperkeratosis	Excision
Maxim et al. (2017) [29]	1, F, 62	5 inflamed enlarging nodules	No data	Right calf	Confined to tattoo	Biopsy: invasive squamous cell carcinoma	Excision
Duan et al. (2016) [19]	1, F, 48	Raised, ruberous, pruritic red ink tattoo area followed by widespread dermatitic eruption on trunk and extremities	1 month	Left foot	Initially confined to tattoo, followed by systemic spread	Patch tests: (−) except for lanolin	Topical clobetasol, intralesional steroids, CO2 laser, systemic steroids (ineffective) Excision (effective)
Joyce et al. (2015) [30]	1, M, 33	Melanoma	3 years	Chest wall	Confined to tattoo	Biopsy: melanoma, CT: (−)	Excision with (+) sentinel node biopsy
Godinho et al. (2015) [31]	1, F, 23	Erythema, local edema, and pruritus	2 years	Right ankle	Confined to tattoo	Biopsy: epithelioid granuloma	Allopurinol: recurrence 2 months after treatment
Kazlouskaya et al. (2015) [32]	2: M, 45; F, 44	Itchy eruption and verrucous lesions; pruritic nodular excoriations and pigment alterations	Many years prior with recoloring 7 months prior; several weeks	Right lateral leg; no data	Confined to tattoos	Biopsy: pseudoepitheliomatous hyperplasia	Topical 5-florouracil; spontaneous resolution
Martin-Calizo et al. (2015) [33]	5: F, 28; F, 24; F, 23; F, 38; no data)	Erythema, inflammation; erosion; ulcer	2 years; 4 months; 1 month; 15 days; >15 years	Right: forearm; foot; ankle; wrist; left ankle	Confined to tattoos	Biopsy: inflammatory reaction with pigment in all specimens; granuloma Culture: (−)	No data
Marchesi et al. (2014) [34]	1, M, 35	Linear reddish, nonulcerated plaques	6 months	Right forearm	Confined to tattoo	Patch tests: (−) Biopsy: lymphoid infiltrate compatible with cutaneous pseudolymphoma Chest X-ray, lab tests: (−)	Excision (effective)
Chapman et al. (2014) [35]	1, M, age not provided	Erythematous plaques	20 years	No data	Confined to tattoos	Patient refused	Spontaneous resolution after neutropenia resolution
Feldstein et al. (2014) [36]	1, F, 29	Erythematous ulcerations with honey-colored crusting	2 weeks	No data	Confined to tattoo	No data	Mupirocin and white petrolatum and 0.1% triamcinolone ointment: (effective)

F—female, M—male, (−)—negative, (+)—positive.

## Data Availability

Not applicable.

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
