# Peer review of "When Body Art Goes Awry—Severe Systemic Allergic Reaction to Red Ink Tattoo Requiring Surgical Treatment"

_ijerph, 2022, doi:10.3390/ijerph191710741_

Round 1

Reviewer 1 Report

Dear authors,

I’ve reviewed the manuscript "When body art goes awry - severe systemic alllergic reaction to red ink tattoo requiring surgical treatment".
I would like to congratulate the authors for providing highly relevant and moreover, clinically useful case report. However, I would like to share my points of critique with you:

To enhance the quality of this case report, i suggest to do a literature serach regarding clinial cases concering red tattoo color. Here, a brief illustrative (case) summary regarding typical clinical symptoms, diagnostics as well as therapy would be sufficient.

Addtionaly, please add an elavulation of the differences / similarities of different tattoo colors and the clinical impact on humans / patients. 

Please add an illustration of a proposed (diagnostic) algorithm based on the literature and this case report.

Reviewer 2 Report

This report is interesting from a practical point of view. However, the manuscript needs serious revision. The presented information about the clinical case is not structured and does not meet the requirements of the Guidelines for Authors.

Line 1 - Replace the selected paper type. This is a Case Report, not an Article.

Line 3 - Delete the dot at the end of the title.

Write the Citation section: Lastname, F.; Lastname, F.; Lastname, F. Title.

Line 10 - In the Abstract, briefly write the relevance of the paper and add the purpose of your case report.

Line 23 - Design the Keywords section (see the published case reports in this journal as an example).

Line 59 - Write all abbreviations and abbreviations in full when first used in the text.

Introduction - Add the purpose of the presented case report after describing the relevance of the problem.

Line 63 - Remove duplicate subtitle. Add a subtitle 2.1. Clinical Presentation

Line 70 - Place Figure 1 and Figure 2 after the first mention in the text.

Add a subsection 2.2. Laboratory Investigations. Write in this section about the methods of laboratory diagnostics and their results in the patient you observed.

Add subsection 2.3. Differential Diagnosis and Clinical Course.

Add subsection 2.4 Treatment and Interventions. Describe in detail what methods of drug and non-drug treatment were used.

Add subsection 2.5 Treatment and Interventions. Describe in detail what methods of drug and non-drug treatment were used. Move the photo (Figure 3) in the perioperative period to this subsection.

Add subsection 2.6 Outcomes. Describe the dynamics and duration of the recovery period. Specify the time when the photo (Figure 4) was received after the operation. Move the photo in the late postoperative period to this subsection.

References - I recommend replacing old publications (1, 3, 11, 12) with publications of the last 5 years.

Lines 268 - 269 - Add publication date, issue number and pages.

Round 2

Reviewer 1 Report

After the revision, the manuscript significantly improved. I congratulate the authors. 

Author Response

Dear Editor and Reviewers,

Thank you for your approval of the revision of our manuscript entitled " When body art goes awry - severe systemic allergic reaction to red ink tattoo requiring surgical treatment.” Again, we would like to thank anonymous Reviewers for their comments and reviews, which helped to improve the manuscript. In this minor revision we addressed all technical comments concerning text formatting /width of the text, paragraphs, location of figures/tables just after their first mentioning/.

Sincerely,

The Authors

Reviewer 2 Report

I thank the authors for the significant improvement of the manuscript. In my opinion, it has become much more attractive to readers. A small technical fix is needed:

1) format the entire text of the article by width;

2) add paragraphs in the text where they are omitted;

3) place all Figures and Tables in the text immediately after the link to them;

4) add the name of the table.

Author Response

(The authors gave the same response as above.)
